# Combination of Bone Marrow Biopsy and Flow Cytometric Analysis: The Prognostically Relevant Central Approach for Detecting Bone Marrow Invasion in Diffuse Large B-Cell Lymphoma

**DOI:** 10.3390/diagnostics11091724

**Published:** 2021-09-20

**Authors:** Haruya Okamoto, Nobuhiko Uoshima, Ayako Muramatsu, Reiko Isa, Takahiro Fujino, Yayoi Matsumura-Kimoto, Taku Tsukamoto, Shinsuke Mizutani, Yuji Shimura, Tsutomu Kobayashi, Eri Kawata, Hitoji Uchiyama, Junya Kuroda

**Affiliations:** 1Division of Hematology and Oncology, Department of Medicine, Kyoto Prefectural University of Medicine, Kyoto 602-8566, Japan; hokamoto@koto.kpu-m.ac.jp (H.O.); adeliae@koto.kpu-m.ac.jp (A.M.); isa-r@koto.kpu-m.ac.jp (R.I.); kfnyy850@koto.kpu-m.ac.jp (T.F.); m-yayoi@koto.kpu-m.ac.jp (Y.M.-K.); ttsuka@koto.kpu-m.ac.jp (T.T.); mizushin@koto.kpu-m.ac.jp (S.M.); yshimura@koto.kpu-m.ac.jp (Y.S.); t-koba@koto.kpu-m.ac.jp (T.K.); 2Department of Hematology, Japanese Red Cross Kyoto Daini Hospital, Kyoto 602-8031, Japan; uoshiman@koto.kpu-m.ac.jp; 3Department of Hematology, Japanese Red Cross Kyoto Daiichi Hospital, Kyoto 605-0981, Japan; huchiyam@koto.kpu-m.ac.jp; 4Department of Hematology, Matsushita Memorial Hospital, Osaka 570-8540, Japan; esato@koto.kpu-m.ac.jp

**Keywords:** malignant lymphoma, diffuse large B-cell lymphoma, bone marrow invasion, bone marrow survey, bone marrow biopsy, bone marrow clot, flow cytometry, smear

## Abstract

Bone marrow (BM) involvement is associated with prognosis in diffuse large B-cell lymphoma (DLBCL), the most prevalent disease subtype of malignant lymphoma. We conducted this multi-institutional retrospective study to investigate the functional association and prognostic values of four BM tests (BM biopsy, BM clot, flow cytometry (FCM), and BM smear). A total of 221 DLBCL patients were enrolled. BM involvement was detected in 17 (7.7%), 16 (7.2%), 27 (12.2%), and 34 (15.4%) patients by BM biopsy, BM clot, FCM, and BM smear, respectively. The consistency between BM biopsy and clot examination was favorable, with a κ coefficient of 0.705, whereas the consistencies among other modalities were poor. In 184 patients treated with the first-line R-CHOP (-like) regimen, BM involvement was associated with shorter progression-free survival (PFS) irrespective of the type of modality for a positive result. Intriguingly, among various single and combinatory modalities, the combination of BM biopsy and FCM had the highest hazard ratio of 3.33 and a c-index of 0.712. In conclusion, our study suggested that the combination of BM biopsy and FCM is the prognostically relevant central approach for BM involvement detection. The other BM examinations also may provide complementary information in clinical settings.

## 1. Introduction

Diffuse large B-cell lymphoma (DLBCL) is the most common subtype of non-Hodgkin lymphomas (NHLs). It accounts for approximately 30% of NHLs and is a clinically and molecularly heterogeneous disease entity [1,2,3]. The common first-line immunochemotherapy for DLBCL is the so-called R-CHOP therapy, which combines the anti-CD20 monoclonal antibody rituximab and the genotoxic agents of cyclophosphamide, doxorubicin, vincristine, and prednisolone. Although R-CHOP or R-CHOP-like therapies generally cure >60% of patients with newly diagnosed DLBCL [4,5,6,7,8], the remaining patients require more intensive, but frequently more toxic, immunochemotherapeutic strategies and novel approaches with chimeric antigen T-cell therapy, bispecific T-cell engager, or antibody-drug conjugates [9,10,11]. To avoid inappropriate treatment for high-risk disease and overtreatment with excess toxicity for standard-risk disease, it is important to predict treatment success or failure with the current standard first-line treatment in patients with DLBCL. In this regard, various prognostic indices of clinical manifestations have been used in daily clinical practice [4,5,6,7]. Furthermore, various molecular classifications have been proposed to understand disease pathophysiology and predict prognosis in DLBCL [1,2,3].

For the accurate evaluation of disease status and risk prediction, systemic investigation of disease involvement using a series of histopathological/cytological assessments and radiological modalities is an essential initial work-up in lymphoma practice. The organ involvement of lymphoma cells not only determines the disease staging, such as by the conventional Ann Arbor disease staging system, but is also strongly associated with the future clinical outcome, including treatment response, predisposition of future central nervous system invasion, and eventual survival outcome. Bone marrow (BM) is a common disease lesion that affects the survival outcome of patients with DLBCL [12,13,14]. Although the histopathological assessment of tissue specimen obtained by BM biopsy has been the gold standard method for detecting the BM involvement of lymphoma cells, various other modalities, such as the histopathological examination of BM clot sample and cytological diagnosis by flow cytometry (FCM) and/or by microscopic examination of BM smear of hematopoietic cells obtained by BM aspiration, have also been applied for the detection of BM involvement in daily clinical practice. Nevertheless, it remains unclear whether BM clot analysis and cytological assessments may replace BM biopsy in detecting BM involvement in DLBCL. Moreover, the prognostic value of each BM test remains unknown.

This study aimed to identify the diagnostic and prognostic values of the histological assessments of BM biopsy specimen and BM clot and cytological assessments via the microscopic examination of the FCM and smear sections of cells obtained by BM aspiration for detecting BM involvement in patients with DLBCL.

## 2. Materials and Methods

### 2.1. Study Design and Patients

This was a multicenter retrospective study conducted on patients with newly diagnosed DLBCL. They were subjected to simultaneous BM tests via four diagnostic modalities: BM biopsy, clot examination (BM clot), microscopic examination (BM smear), and FCM, between 2012 and 2018 at three independent institutes belonging to the Kyoto Clinical Hematology Study Group. We collected clinical data regarding the results of the four BM tests and survival using case report forms. We then analyzed the positive rate of each diagnostic modality and the correlation among different modalities. We also evaluated the prognostic significance of the positive result with each modality in patients treated with R-CHOP or R-CHOP-like regimen.

### 2.2. Detection and Definition of BM Involvement

The BM involvement of lymphoma cells was analyzed via four different modalities: two histological assessments and two cytological assessments. BM trephine biopsy and aspiration were performed from the posterior iliac crest in all patients. BM biopsy specimen and BM aspirate clot samples were subjected to hematoxylin-eosin staining and to immunohistochemical staining for CD5, CD10, CD20, CD79a, BCL6, and MUM-1, as necessary. Then, biopsied and aspirate samples were histologically evaluated as BM biopsy and BM clot, respectively, by hematopathologists. For the cytological detection of clonal lymphoma cells by FCM analysis, BM nucleated cells from aspirate specimen were stained using a two-color direct immunofluorescence technique using several monoclonal antibodies conjugated with either Fluorescein Isothiocyanate, Phycoerythrin, or Allophycocyanin. Antigens examined included CD2, CD3, CD4, CD7, CD8, CD10, CD11c, CD16, CD19, CD20, CD23, CD25, CD34, CD45, CD56, kappa light chain, and lambda light chain. The presence of light chain restriction in B lymphoid cells was regarded as positive BM invasion [15,16,17,18,19,20]. BM aspirate smear specimen was subjected to Wright–Giemsa staining and 500 BM nucleated cells were cytologically evaluated to detect abnormal lymphoid cells under an optical microscope. In the smear section, lymphoma involvement was considered to be positive based on the presence of at least 1% of lymphoma cells in all nucleated cells.

### 2.3. Statistical Analysis

Progression-free survival (PFS) was defined as the time from the date of treatment initiation for DLBCL to the first relapse or progression, death from any cause, or the last follow-up. Overall survival (OS) was defined as the time from the date of treatment initiation for DLBCL to death from any cause or the date of the last follow-up. OS and PFS were estimated using the Kaplan–Meier method, and differences between the groups were compared using the log-rank test. Cox proportional-hazard models were constructed to evaluate the effect of BM involvement detected in BM biopsied specimen, in clot specimen, and via cytological examination of FCM analysis and smear sample. Multivariate analysis was conducted by adjusting the following risk factors included in the International Prognostic Index (IPI): age, lactate dehydrogenase (LDH), performance status (PS) (2–4 versus 0–1) based on the criteria defined by the Eastern Cooperative Oncology Group (ECOG), number of extranodal sites (≥2 versus 0–1), and disease stage according to the Ann Arbor staging system (III–IV versus I–II). The concordance index (c-index), which is the conditional probability that the patients with a longer event time were estimated at a lower risk, was used to measure the separation of survival distributions among different risk groups under the adjustment of the risk factors included in the IPI. The *p* values ≤ 0.05 were considered to be statistically significant. All statistical analyses were conducted using EZR 1.42 [21].

## 3. Results

### 3.1. Patients

The data of 221 patients were analyzed in this study. As presented in Table 1, the median age of the patients was 72 (range: 26–97) years, and 135 (61.1%) were men. According to the revised-IPI (R-IPI), 12 (5.4%), 92 (41.6%), and 117 (52.9%) patients exhibited very good, good, and poor-risk, respectively, indicating that the risk classification pattern according to the R-IPI generally showed common distribution in our study cohort [4,6]. Of the 221 patients, 184 were initially treated with R-CHOP or R-CHOP-like regimen.

### 3.2. BM Lymphoma Invasion Detected by Four Different Modalities

As presented in Figure 1A and Table 2A, BM involvement was detected by at least one of the four modalities in 55 (24.9%) of the 221 patients analyzed in this study. Precisely, BM involvement was detected in 17 (7.7%) and 16 (7.3%) patients through the histological assessments of biopsied BM specimen and BM clot samples, respectively. In 27 (12.2%) and 34 (15.4%) patients, it was detected through the cytological assessments of BM aspirate sample using FCM analysis and through the microscopic examination of smear specimen, respectively. Therefore, the sensitivities for the detection of BM involvement were generally equivalent between two modalities for histological assessments and between two modalities for cytological assessments. Furthermore, the sensitivities for detecting BM involvement were not statistically significantly different between two cytological and two histological modalities, although the former detected more BM involvement than the latter. Discrepancies were consistently observed between the positive and negative results among all the four modalities employed in this study (Figure 1A and Table 2A). Among the 221 patients, only four were diagnosed with BM involvement by all four modalities, whereas 7, 13, and 31 patients were considered positive for BM involvement by three, two, and one of the four modalities, respectively. Moreover, BM involvement was determined to be negative by all four methods in 166 patients. We investigated the coefficiencies among the different modalities, which demonstrated that the results obtained by BM biopsied specimen and BM clots indicated favorable consistency with a κ coefficiency of 0.705. However, no favorable consistency was observed between any of the other combinations, even between two cytological assessments (Table 2A).

### 3.3. Prognostic Significance of the Detection of BM Involvement by Different Modalities

#### 3.3.1. Patients

To investigate the clinical importance, we next examined the prognostic significance of various modalities for BM involvement. For this purpose, we focused on the 184 patients who were almost uniformly treated with R-CHOP or R-CHOP-like regimens. In this subcohort of 184 patients, BM involvement was detected by at least one of the four modalities in 42 (22.8%) patients. Precisely, BM involvement was detected in 14 (7.6%) and 12 (6.5%) patients in biopsied BM specimen and BM clot samples, respectively, and in 18 (9.8%) and 26 (14.1%) patients by FCM analysis and aspirate smear examination, respectively (Figure 1B and Table 2B). Again, a favorable coefficiency was observed only between the investigation of BM biopsied specimen and BM clot samples. In this subcohort, the PFS and OS at 2 years were 67.5% and 75.8%, respectively, with a median follow-up period of 31 months (Figure 2).

#### 3.3.2. Prognostic Impact of the Detection of BM Involvement by BM Biopsy

As a gold standard approach for BM analysis, we first analyzed the clinical impact of the detection of BM involvement by BM biopsy. Our result indicated that BM involvement detected by BM biopsy was significantly associated with PS, increased serum level of LDH, and more advanced disease stage. Moreover, BM involvement was significantly associated with anemia and decreased platelet counts in the peripheral blood (Table 3). Intriguingly, BM involvement detected by BM biopsy had a significant impact on patient survival, with the 2-year PFS and 2-year OS being 19.6% (95% CI; 3.1–46.5%) and 23.8% (95% CI; 5.8–48.5%), respectively, whereas in patients with no detectable BM invasion by BM biopsy being 71.4% (95% confidence interval (CI); 63.6–77.9%) and 80.4% (95% CI; 73.3–85.8%), respectively (*p* < 0.001) (Figure 3A,B).

#### 3.3.3. Optimal Detection Method of BM Involvement for Prediction of Prognosis

We next investigated the single modality or any combination of BM tests that exerts the most powerful prognostic impact through the detection of BM involvement. First, the Cox proportional-hazard model demonstrated that the detection of BM involvement not only by single BM test but also by any combination of more than two BM tests was significantly associated with the increased risk of shorter PFS in both univariate and multivariate analyses (Table 4). Regarding OS, although the univariate analysis demonstrated that the detection of BM involvement was associated with poor outcome irrespective of the type or combinatory pattern of BM tests, this was not always true in the multivariate analysis (Appendix A). BM biopsy revealed the highest hazard ratio (HR), whereas FCM showed the highest c-index for both PFS and OS as a single modality with the detection of BM involvement. In addition, the detection of BM involvement by the combination of BM biopsy and FCM showed the highest HR and c-index throughout all single and combinatory BM tests for both PFS and OS (Table 4 and Appendix A). Accordingly, the 2-year PFS and 2-year OS were 75.8% (95% CI; 67.9–77.9%) and 82.2% (95% CI; 74.9–87.5%), respectively, in patients with no detectable BM invasion by BM biopsy and/or FCM (*p* < 0.001) and 19.6% (95% CI; 3.1–46.5%) and 42.2% (95% CI; 23.0–60.2%), respectively, in patients with detectable BM invasion by BM biopsy and/or FCM (*p* < 0.001) (Figure 3C,D).

## 4. Discussion

We conducted this multi-institutional retrospective study to comparatively investigate the sensitivities of four BM tests (two histological and two cytological examinations), the consistency among the different modalities, and their prognostic significance in the real-world setting.

Regarding the sensitivity for detecting BM involvement, our results showed a trend of higher positive rates with cytological assessments than with histological assessments. This tendency was consistent with those in previous studies, indicating the superior sensitivity of FCM than of BM biopsy in detecting BM involvement [16,17]. Such a different sensitivity may reflect the difficulties in the histological detection of lymphoma cell invasion in a tissue sample, which could be influenced by the proportion of lymphoma cells and the form of invasion in BM.

Regarding the consistency of the different modalities, only BM biopsy and clot examination showed a favorable κ coefficiency. This result was somewhat surprising, as the two samples are different in their origins, i.e., BM biopsy being a tissue sample, whereas BM clot being from a bone marrow aspirate. However, our result was consistent with previous studies that indicated that BM clot assessment with an adequate amount of specimen material allows the same histological evaluation as BM biopsy. Moreover, BM clot is more useful in immunohistochemical evaluations, as BM clot may present better antigenic expression, mainly because it does not need a decalcification procedure which is inevitable in BM biopsy [22,23]. BM biopsy is a safe technique, although it is rather more invasive than BM aspiration [24,25,26]. Considering our results indicated the higher frequency of low platelet count and poorer PS in patients with BM involvement detected by BM biopsy, the histological examination of BM clot samples obtained by aspiration could be the alternative to BM biopsy, especially in cases with severe bleeding tendency, high risk for bone fracture, or frail status. Contrarily, despite the similar sensitivities of BM smear examination and FCM analysis, the consistency was insufficient between the two modalities. In fact, in 51 patients evaluated as having a positive result by either FCM or BM smear, only 10 (19.6%) were simultaneously evaluated as positive by these two cytological diagnostic modalities. This result may reflect the technical difference between two modalities and raises the question about the differential clinical significance between two modalities. One of the most attractive characteristics of FCM is its ability to detect B-cell clonality in BM cells, which leads to a straightforward and objective diagnosis for lymphoma invasion in BM. However, in lymphoma practice, BM smear analysis is unavoidable as this method potentially detects the invasion of lymphoma cells and the presence of other hematologic abnormalities, such as the coexistence of myelodysplasia and hemophagocytosis.

Given the different sensitivities and the coefficiencies among the four modalities investigated in this study, we finally evaluated the best approach for detecting BM involvement from the viewpoint of prognostic impact. As a single modality, although the detection of BM involvement by biopsy showed the highest HR, the c-index was the highest with FCM for PFS in our cohort treated with R-CHOP (-like) first-line treatment. Perhaps, due to these prognostic impacts of two modalities, the combination of BM biopsy and FCM showed the highest HR and c-index for PFS and even OS. Intriguingly, in 37 patients with BM involvement detected by the combination of BM biopsy and FCM, only seven (18.9%) were simultaneously found to have a positive result by these two modalities. Based on this finding, we speculate the complementary role between the two modalities. Although the prognostic association between BM involvement detection by biopsy and FCM has been controversial [18,19,20,27], our result indicated the combination of BM biopsy and FCM as the central component in BM analysis. BM smear and clot examinations are also useful modalities, as they may provide additional information and play the alternative role in some clinical situations.

In recent years, ^18^F-fluorodeoxyglucose positron emission tomography and computed tomography (^18^F-FDG PET-CT) have been reported to possess high BM involvement detection sensitivity and have been proposed to replace BM biopsy [28,29]. However, other studies have suggested that the sensitivity of ^18^F-FDG PET-CT is inferior to that of the histological assessments by BM biopsy [30,31,32]. Moreover, several studies have demonstrated that the prognosis of patients with DLBCL with BM invasion was inferior with detection by biopsy compared with detection by ^18^F-FDG PET-CT [33,34,35]. In this study, unfortunately, ^18^F-FDG PET-CT was performed in 154 of the 184 patients treated with the R-CHOP-like regimen, which may induce patient selection bias to evaluate prognostic value. In fact, patients who did not receive ^18^F-PET-CT had significantly poor PS compared with those who underwent PET-CT in our cohort (data not shown). Perhaps, at least partially due to such bias, we found the trend of poor prognosis only in patients with BM invasion detected by PET without a statistically significant difference (Appendix A). Because ^18^F-FDG PET-CT is potent in systemically surveying BM involvement, PET/CT-guided BM biopsy may be more diagnostically reliable than the current routine BM trephine biopsy. However, it may also be sometimes more invasive for patients.

Finally, one of the conceivable limitations in this study is the lack of consideration for or observation of concordant BM disease defined by the involved BM area consisting of mostly large noncleaved lymphoma cells and discordant BM disease defined by the involved BM consisting of mostly small low-grade lymphoma cells in DLBCL [36,37,38]. It has been reported that the subclassification of invaded lymphoma cells into two morphologically distinct types, i.e., concordant and discordant types, is associated with prognosis [36,37,38]. However, we could not evaluate this point because pathologists reported no discordant BM involvement with BM biopsy or BM clot assessment in our study. Considering discordant clonal lymphoid cells occasionally reflects the incidental co-occurrence of indolent clonal lymphoid proliferation with DLBCL [36,37], one of the conceivable reasons which underlay the discrepancy between frequent discordant BM involvement in previous reports and the absence of discordant BM disease in our cohort may be the distinct racial impact on the frequency of indolent B cell neoplasms, such as follicular lymphoma, chronic lymphoid leukemia (CLL) and its precursor, monoclonal B-cell lymphocytosis (MBL). For instance, the incidence of CLL was about 5–10 folds less in Asians than in Caucasians [39,40], and this was also the case with MBL [41]. Therefore, it was difficult to identify a rare patient with discordant BM disease and investigate its clinical value in our cohort with a limited number of patients. A future study with a larger number of DLBCL patients is expected to elucidate the association between the type of concordant/discordant and FCM phenotype in Asia. In addition, further interesting research topics may include the investigation of the prognostic impact of the histological pattern of lymphoma cells and the situation of surrounding immune cells in the tumor microenvironment, their associations with the type of concordant/discordant disease, and the molecular classification of tumor cells that are functionally active in tumorous lesions [42,43].

## 5. Conclusions

The present study indicated the association between BM involvement detected by any modality and poor prognosis and suggested that the screening of BM involvement by biopsy and/or FCM provides the most prognostically important information in DLBCL. The BM clot analysis study could be a better alternative to BM biopsy if required, and BM smear examination and FCM may play their peculiar roles in terms of providing different aspects of disease information.

## Figures and Tables

**Figure 1 diagnostics-11-01724-f001:**
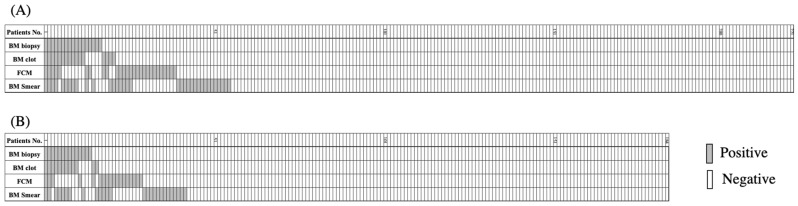
Results of bone marrow (BM) examination via four modalities. (**A**) all patients and (**B**) patients initially treated with R-CHOP or R-CHOP-like regimen. FCM, flow cytometry; smear, microscopic examination of BM aspirate smear section. Gray boxes indicate patients with a positive result, and white boxes indicate patients with a negative result.

**Figure 2 diagnostics-11-01724-f002:**
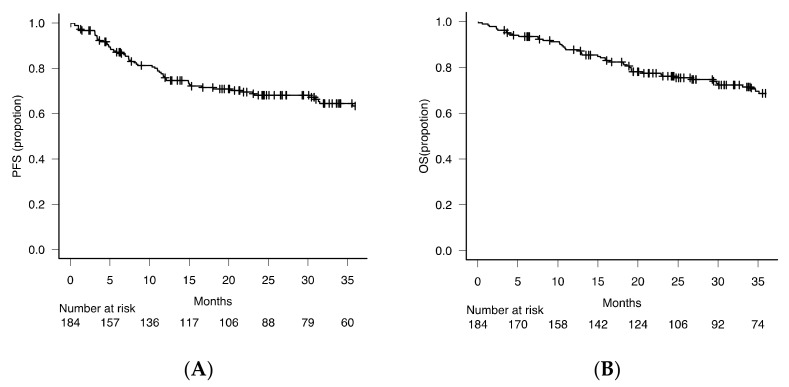
Survival curves. Kaplan–Meier curves for (**A**) progression-free survival (PFS) and (**B**) overall survival (OS) of 184 patients with DLBCL initially treated with R-CHOP or R-CHOP-like regimen and analyzed in this study. BM, bone marrow; DLBCL, diffuse large B cell lymphoma; OS, overall survival; PFS, progression free survival.

**Figure 3 diagnostics-11-01724-f003:**
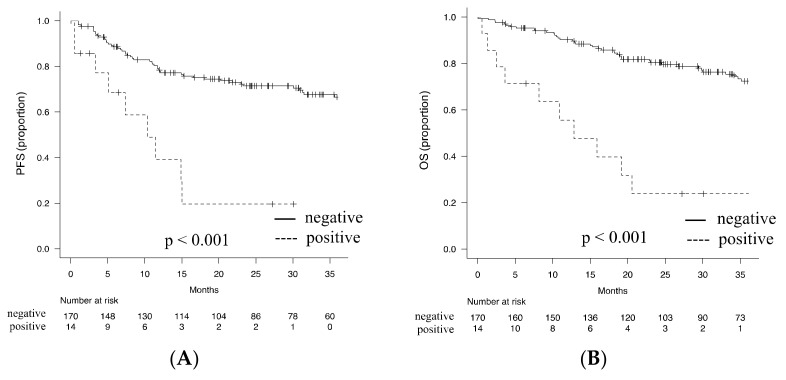
Survival curves in association with BM involvement. A total of 184 patients with DLBCL initially treated with R-CHOP or R-CHOP-like regimen were analyzed. (**A**,**B**). PFS (**A**) and OS (**B**) according to the detection of BM involvement by BM biopsy. (**C**,**D**). PFS (**C**) and OS (**B**) according to the detection of BM involvement by BM biopsy and/or FCM.BM, bone marrow; DLBCL, diffuse large B cell lymphoma; OS, overall survival; PFS, progression free survival.

**Table 1 diagnostics-11-01724-t001:** Patient background.

Characteristics	
Total Number of Patients, *n*	221
Age, median (range)	72 (26–97)
Gender, Male, *n* (%)	135 (61.1)
ECOG PS 2–4, *n* (%)	68 (31.2)
LDH > ULN, *n* (%)	131 (59.3)
Ann Arbor Stage III/IV, *n* (%)	127 (57.5)
Extranodal site 2 or more, *n* (%)	63 (28.5)
R-IPI, *n* (%)	
Very good	12 (5.4)
Good	92 (41.6)
Poor	117 (52.9)
Treatment regimen, *n* (%)	
R-CHOP (-like) regimen	184 (83.3)
Intensive chemotherapy other than R-CHOP	17 (7.7)
Upfront autologous stem cell transplantation	13 (5.9)
Palliative care	7 (3.2)

ECOG, Eastern Cooperative Oncology Group; LDH, serum lactate dehydrogenase; PS, performance status; R-CHOP, rituximab plus CHOP; R-IPI, revised International Prognostic Index; ULN, upper normal limit.

**Table 2 diagnostics-11-01724-t002:** Detection rates of bone marrow invasion of lymphoma cells by different modalities and the interexamination coefficiency.

A. All Patients (*n* = 221)											
Modality			BM Biopsy			BM Clot			FCM		
	Result/*n* (%)		−	+	κ	−	+	κ	−	+	κ
			204 (92.3)	17 (7.7)		205 (92.7)	16 (7.3)		194 (87.8)	27 (12.2)	
BM smear	−	187 (84.6)	181 (81.9)	6 (2.7)	0.366	182 (82.4)	5 (2.3)	0.379	170 (76.9)	17 (7.7)	0.222
	+	34 (15.4)	23 (10.4)	11 (5)		23 (10.4)	11 (5)		24 (10.9)	10 (4.5)	
FCM	−	194 (87.8)	184 (83.3)	10 (4.5)	0.247	185 (83.7)	9 (4.1)	0.258			
	+	27 (12.2)	20 (9.0)	7 (3.2)		20 (9.0)	7 (3.2)				
BM clot	−	205 (92.7)	200 (90.5)	5 (2.3)	0.705						
	+	16 (7.3)	4 (1.8)	12 (5.4)							
**B. Patients Initially Treated with R-CHOP or R-CHOP-Like Regimen (*n* = 184)**											
**Modality**			**BM Biopsy**			**BM Clot**			**FCM**		
	Result/*n* (%)		−	+	κ	−	+	κ	−	+	κ
			170 (92.4)	14 (7.6)		172 (93.5)	12 (6.5)		166 (90.2)	18 (9.8)	
BM smear	−	158 (85.9)	152 (82.6)	6 (3.3)	0.334	154 (83.7)	4 (2.2)	0.364	146 (79.3)	12 (6.5)	0.178
	+	26 (14.1)	18 (9.8)	8 (4.3)		18 (9.8)	8 (4.3)		20 (10.9)	6 (3.3)	
FCM	−	166 (90.2)	156 (84.8)	10 (5.4)	0.180	158 (85.9)	8 (4.3)	0.204			
	+	18 (9.8)	14 (7.6)	4 (2.2)		14 (7.6)	4 (2.2)				
BM clot	−	172 (93.5)	168 (91.3)	4 (2.2)	0.752						
	+	12 (6.5)	2 (1.1)	10 (5.4)							

BM, bone marrow; BM clot, histological examination of BM clot; FCM, flow cytometry; κ, κ coefficient; −, negative; +, positive.

**Table 3 diagnostics-11-01724-t003:** Clinical characteristics of 184 patients treated with R-CHOP (-like) regimen in association with BM involvement detected by BM biopsy.

Clinical Characteristics	BM Involvement Diagnosed by BM Biopsy		
	Negative	Positive	*p* Value
Number of patients, *n*	170	14	-
Age, median (range)	73 (28–89)	76 (43–89)	0.393
Gender, Male, *n* (%)	105 (61.8)	7 (50.0)	0.405
ECOG PS 2–4, *n* (%)	40 (23.8)	11 (78.6)	<0.001
LDH > ULN, *n* (%)	90 (52.9)	13 (92.4)	0.004
Disease stage III, IV, *n* (%)	84 (49.4)	14 (100.0)	<0.001
Extranodal site 2 or more, *n* (%)	40 (23.5)	4 (28.6)	0.745
Leukocyte count, median (×10^9^/L) (range)	6.14 (1.80–5.87)	5.50 (1.10–2.48)	0.497
Hemoglobin level, median (g/dL) (range)	12.4 (5.2–16.2)	9.9 (6.7–14.5)	0.011
Platelet, median (×10^9^/L) (range)	217.0 (180.0–763.0)	105.0 (80.0–215.0)	<0.001
R-IPI, *n* (%)			0.003
Very good	12 (7.1)	0 (0.0)	
Good	80 (47.1)	1 (7.1)	
Poor	78 (45.9)	13 (92.9)	

ECOG, Eastern Cooperative Oncology Group; LDH, serum lactate dehydrogenase; PS, performance status; R-CHOP, rituximab plus CHOP; R-IPI, revised International Prognostic Index; ULN, upper normal limit.

**Table 4 diagnostics-11-01724-t004:** Impact of the detection of BM involvement by single or combinatory modality on progression-free survival.

		Univariate Analysis			Multivariate Analysis *			
Modality	Result (*n*)	HR	95% CI	*p*	HR	95% CI	*p*	c-Index
BM biopsy	− (170)	1	−	−	1	−	−	0.684
	+ (14)	4.29	2.08–8.83	<0.001	2.87	1.27–6.50	0.011	
BM clot	− (172)	1	−	−	1	−	−	0.688
	+ (12)	4.37	2.14–8.92	<0.001	2.69	1.24–5.83	0.012	
FCM	− (166)	1	−	−	1	−	−	0.703
	+ (18)	3.06	1.63–5.78	<0.001	2.75	1.44–5.24	0.002	
BM smear	− (158)	1	−	−	1	−	−	0.702
	+ (26)	3.50	2.02–6.08	<0.001	2.46	1.36–4.44	0.003	
BM biopsy and/or clot	− (186)	1	−	−	1	−	−	0.69
	+ (16)	3.75	1.89–7.43	<0.001	2.42	1.16–5.07	0.019	
BM biopsy and/or FCM	− (156)	1	−	−	1	−	−	0.712
	+ (28)	3.94	2.27–6.82	<0.001	3.33	1.85–6.00	<0.001	
BM biopsy and/or BM smear	− (152)	1	−	−	1	−	−	0.695
	+ (32)	3.08	1.79–5.31	<0.001	2.13	1.19–3.83	0.011	
BM clot and/or FCM	− (158)	1	−	−	1	−	−	0.705
	+ (26)	3.66	2.10–6.36	<0.001	2.89	1.62–5.17	<0.001	
BM clot and/or BM smear	− (154)	1	−	−	1	−	−	0.707
	+ (30)	3.60	2.11–6.14	<0.001	2.54	1.42–4.55	0.002	
FCM and/or BM smear	− (146)	1	−	−	1	−	−	0.708
	+ (38)	3.29	1.97– 5.50	<0.001	2.50	1.45–4.30	<0.001	
BM biopsy, clot, and/or FCM	− (155)	1	−	−	1	−	−	0.705
	+ (29)	3.52	2.04–6.09	<0.001	2.82	1.58–5.01	<0.001	
BM biopsy, clot, and/or BM smear	− (151)	1	−	−	1	−	−	0.704
	+ (33)	3.31	1.94– 5.66	<0.001	2.32	1.30–4.14	<0.001	
BM biopsy, FCM, and/or BM smear	− (142)	1	−	−	1	−	−	0.706
	+ (42)	3.19	1.92–5.31	<0.001	2.37	1.37–4.08	0.002	
BM clot, FCM, and/or BM smear	− (144)	1	−	−	1	−	−	0.707
	+ (40)	3.39	2.03–5.64	<0.001	2.52	1.47–4.34	<0.001	
BM biopsy, clot, FCM, and/or BM smear	− (142)	1	−	−	1	−	−	0.706
	+ (42)	3.19	1.92–5.31	<0.001	2.37	1.37–4.08	0.002	

BM, bone marrow; CI, confidence interval; HR, hazard ratio; −, negative; +, positive. * adjusting for IPI factors; age, LDH, PS > 1, stage >II, extranodal site.

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
