# Peer review of "Combination of Bone Marrow Biopsy and Flow Cytometric Analysis: The Prognostically Relevant Central Approach for Detecting Bone Marrow Invasion in Diffuse Large B-Cell Lymphoma"

_diagnostics, 2021, doi:10.3390/diagnostics11091724_

Round 1

Reviewer 1 Report

The manuscript by Okamoto H. and coll. "Combination of bone marrow biopsy and flow cytometric analysis: The prognostically relevant central approach for detecting bone marrow invasion in diffuse large B-cell lymphoma" analyzes the impact of four different approaches to the study of bone marrow invasion in diffuse large B-cell lymphoma (DLBCL) on the clinical behavior and survival of patients affected by this type of lymphoma. The results obtained suggest a worse prognosis in case of bone marrow involvement regardless of the method used to detect it and the highest hazard ratio of the combination of bone marrow biopsy and flow cytometry.

One of the main issues concerns materials and methods.

The description of the methods used for the analysis of BM samples is very essential. The immunohistochemical stains used should be specified, as well as the possible use of markers to define the cell of origin of DLBCL (germinal center-like vs activated-like). Flow cytometry analysis should be detailed (number of parameters, fluorochromes used, etc.) and the source of the samples (presumably BM aspirate) should be specified.

Editing of tables and figures must be carefully controlled (e.g., in Table 2 both tables are identified as "A". In the second column of the same tables the "+" and "-" are reversed).

A point requiring to be reevaluated is concerning the four methods used for the BM analysis. The BM clot is assimilated to BM trephine biopsy. While this is true for the analytical method used (histology and immunohistochemistry), the two samples are very different, the BM biopsy being a "true" histological sample (tissue), whereas the BM is obtained from a bone marrow aspirate. The origin of the sample should override the analytical method used, so the BM clot should be more similar (if not identical) to the two "cytologic" samples (FC and smear) than the BM biopsy. From this point of view, the greater consistency of the bone marrow biopsy and clot compared with the other two sample types is surprising.

The analysis of concordant and discordant BM infiltrates is a critical point especially with regard to flow cytometry. A recent article analyzed this point in detail, so a reanalysis of the data considering this aspect as well could increase the significance of the research.

Alonso-Alvarez S et al. Biological Features and Prognostic Impact of Bone Marrow Infiltration in Patients with Diffuse Large B-cell Lymphoma. Cancers 2020, 12, 474; doi:10.3390/cancers12020474.

Author Response

Dear reviewer 1

We greatly appreciate you for kindly giving us thoughtful comments and for having us the opportunity for the submission of the revised article. We would like to make replies to comment as below.

We also would like to make corrections in red ink with comments in the revised article instead of using the Trach Changes function, because the version of MS Word is basically Japanese mode, therefore, the Track Changes function is also shown in Japanese language which should be inconvenient for editors and reviewers. We would like to apologize for this inconvenience.

In reply to reviewer 1

First of all, we would like to thank reviewer 1 for reviewing our article precisely.

Comment 1)

The immunohistochemical stains used should be specified, as well as the possible use of markers to define the cell of origin of DLBCL (germinal center-like vs activated-like). Flow cytometry analysis should be detailed (number of parameters, fluorochromes used, etc.) and the source of the samples (presumably BM aspirate) should be specified

Reply to comment 1)

We greatly appreciate this important comment. We would like to add the information about markers used in immunohistochemical assessment and more detailed information about flow cytometric analysis in Materials and Methods as follows on pages 2 and 3; “BM biopsy specimen and BM aspirate clot samples were subjected to hematoxylin–eosin staining and to immunohistochemical staining for CD5, CD10, CD20, CD79a, BCL6, and MUM-1, as necessary. Then, biopsied and aspirate samples were histologically evaluated as BM biopsy and BM clot, respectively, by hematopathologists.” and “For the cytological detection of clonal lymphoma cells by FCM analysis, BM nucleated cells from aspirate specimen were stained using a two-color direct immunofluorescence technique using several monoclonal antibodies conjugated with either Fluorescein Isothiocyanate, Phycoerythrin, or Allophycocyanin. Antigens examined included CD2, CD3, CD4, CD7, CD8, CD10, CD11c, CD16, CD19, CD20, CD23, CD25, CD34, CD45, CD56, kappa light chain, and lambda light chain. The presence of light chain restriction in B lymphoid cells was regarded as positive BM invasion.”

Comment 2)

Editing of tables and figures must be carefully controlled (e.g., in Table 2 both tables are identified as "A". In the second column of the same tables the "+" and "-" are reversed).

Reply to comment 2)

We appreciate your great help. We would like to amend our Tables and Figure.

Comment 3)

A point requiring to be reevaluated is concerning the four methods used for the BM analysis. The greater consistency of the bone marrow biopsy and clot compared with the other two sample types is surprising.

Reply to comment 3)

We appreciate the thoughtful comment by the reviewer. We agree with your consideration, while we at the same time learned that, in addition to our own result, the utility of BM aspirate clot as an alternative to BM biopsy from previous literatures (new references No.22 and 23). Those studies indicated that BM clot assessment with an adequate amount of specimen material allows the same kind of histological evaluation as BM biopsy and provides a wide range of information. BM clot is especially valuable in case BM biopsy material is small or inadequate for the diagnosis. The hematopoietic tissue is enriched in the clot exam, allowing adequate cuts, utilization of several histochemical staining, and/or immunophenotypical evaluation and/or molecular analysis. When compared to BM biopsy, BM clot may present better antigenic expression in IHC techniques, mainly because it does not need decalcification procedure which is inevitable in BM biopsy. In our revised article, we would like to add this in Discussion section, and also would like to add two new references as below.

  1. Miranda, R.N.; Mark, H.F.; Medeiros, L.J. Fluorescent in situ hybridization in routinely processed bone marrow aspirate clot and core biopsy sections. Am J Pathol. 1994, 145, 1309-1314.

  1. Ong, M.G.; Lowery-Nordberg, M.; Pillarisetti, S.; Veillon, D.; Cotelingam, J. Maximizing the diagnostic yield from bone marrow aspirate material using the cell block technique on clot sections. Lab Med. 2015, 46, e24-27.

Comment 4)

The analysis of concordant and discordant BM infiltrates is a critical point especially with regard to flow cytometry. A recent article analyzed this point in detail, so a reanalysis of the data considering this aspect as well could increase the significance of the research.

Alonso-Alvarez S et al. Biological Features and Prognostic Impact of Bone Marrow Infiltration in Patients with Diffuse Large B-cell Lymphoma. Cancers 2020, 12, 474; doi:10.3390/cancers12020474.

Reply to comment4)

We agree with the reviewer’s consideration, and also greatly appreciate for introducing the excellent article. However, in our study, no discordant BM involvement was reported by pathologists with BM biopsy or BM clot assessment. Considering discordant clonal lymphoid cells occasionally reflects the incidental co-occurrence of indolent clonal lymphoid proliferation with DLBCL, one of the conceivable reasons which underlay the discrepancy between frequent discordant BM involvement in previous reports and the absence of discordant BM disease in our cohort may be the distinct racial impact on the frequency of indolent B cell neoplasms, such as follicular lymphoma, chronic lymphoid leukemia (CLL) and its precursor, monoclonal B-cell lymphocytosis (MBL). For instance, the incidence of CLL was about 5-10 folds less in Asians than in European descent and this was also the case with MBL. Therefore, it was difficult to identify a rare patient with discordant BM disease and investigate its clinical value in our cohort with limited number of patients, and future study with larger number of DLBCL patients is expected to elucidate the association between the type of concordant/discordant and FCM phenotype in Asia. We would like to add this consideration in Discussion section. For this discussion, we also would like to add four new references as below.

  1. Alonso-Álvarez, S.; Alcoceba, M.; García-Álvarez, M.; Blanco, O.; Rodríguez, M.; Baile, M.; Caballero, J.C.; Dávila, J.; Vidriales, M.B.; Esteban, C.; et al. Biological Features and Prognostic Impact of Bone Marrow Infiltration in Patients with Diffuse Large B-Cell Lymphoma. Cancers 2020, 12, E474, doi:10.3390/cancers12020474.
  2. Morton, L.M.; Wang, S.S.; Devesa, S.S.; Hartge, P.; Weisenburger, D.D.; Linet, M.S. Lymphoma Incidence Patterns by WHO Subtype in the United States, 1992-2001. Blood 2006, 107, 265–276, doi:10.1182/blood-2005-06-2508.
  3. Yang, S.-M.; Li, J.-Y.; Gale, R.P.; Huang, X.-J. The Mystery of Chronic Lymphocytic Leukemia (CLL): Why Is It Absent in Asians and What Does This Tell Us about Etiology, Pathogenesis and Biology? Blood Rev. 2015, 29, 205–213, doi:10.1016/j.blre.2014.12.001.
  4. Yoo, I.Y,; Bang, S.H.; Lim, D.J.; Kim, S.J.; Kim, K. Kim, H.J.; Kim, S.H.; Cho, D. Prevalence and Immunophenotypic Characteristics of Monoclonal B-Cell Lymphocytosis in Healthy Korean Individuals With Lymphocytosis. Ann Lab Med. 2020, 40, 409-413. doi: 10.3343/alm.2020.40.5.409.

We again thank thoughtful and excellent suggestions and comments by the expert reviewer. Thanks to the great comments by expert reviewers, we believe that the revised article becomes much better than the original version.

Your consideration will be much appreciated, and I look forward to hearing from you.

Sincerely yours,

 Junya Kuroda MD., Ph.D.

 Professor and Chair

Division of Hematology and Oncology,

Kyoto Prefectural University of Medicine

465, Kajii-cho, Kamigyo-ku, Kyoto, Japan

E-email: [email protected]

Reviewer 2 Report

The retrospective study on functional association and prognosis of BM tests in DLBCL is of great interest. The authors have convincingly shown that BM biopsy and FCM approach may help in delineating BM involvement.

Author Response

Dear Reviewer 2

We greatly appreciate you for kindly giving us encouraging comments and for having us the opportunity for the submission of the revised article. We are also delighted that reviewers found some merits in our report. We tried our best to reply to the comments by reviewers. We would like to make replies to comment as below.

We also would like to make corrections in red ink with comments in the revised article instead of using the Trach Changes function, because the version of MS Word is basically Japanese mode, therefore, the Track Changes function is also shown in Japanese language which should be inconvenient for editors and reviewers. We would like to apologize for this inconvenience.

In reply to reviewer 2

Reviewers' comments:

The retrospective study on functional association and prognosis of BM tests in DLBCL is of great interest. The authors have convincingly shown that BM biopsy and FCM approach may help in delineating BM involvement.

In reply to review 2:

We greatly appreciate reviewer 2 for kind comments.

We again thank thoughtful and excellent suggestions and comments by two expert reviewers. Thanks to the great comments by expert reviewers, we believe that the revised article becomes much better than the original version.

Your consideration will be much appreciated, and I look forward to hearing from you.

Sincerely yours,

Junya Kuroda MD., Ph.D.

Professor and Chair

Division of Hematology and Oncology,

Kyoto Prefectural University of Medicine

465, Kajii-cho, Kamigyo-ku, Kyoto, Japan

E-email: [email protected]

Round 2

Reviewer 1 Report

The Authors have responded appropriately to the comments raised.

No further corrections are necessary.